# Daily Lifestyle Modifications to Improve Quality of Life and Survival in Glioblastoma: A Review

**DOI:** 10.3390/brainsci11050533

**Published:** 2021-04-23

**Authors:** Sarah Travers, N. Scott Litofsky

**Affiliations:** Division of Neurosurgery, University of Missouri School of Medicine, Columbia, MO 65212, USA; traverss@health.missouri.edu

**Keywords:** survival in glioblastoma, dietary restriction in glioblastoma, cannabis use in glioblastoma, supplementation in glioblastoma, glioblastoma health modifications

## Abstract

Survival in glioblastoma remains poor despite advancements in standard-of-care treatment. Some patients wish to take a more active role in their cancer treatment by adopting daily lifestyle changes to improve their quality of life or overall survival. We review the available literature through PubMed and Google Scholar to identify laboratory animal studies, human studies, and ongoing clinical trials. We discuss which health habits patients adopt and which have the most promise in glioblastoma. While results of clinical trials available on these topics are limited, dietary restrictions, exercise, use of supplements and cannabis, and smoking cessation all show some benefit in the comprehensive treatment of glioblastoma. Marital status also has an impact on survival. Further clinical trials combining standard treatments with lifestyle modifications are necessary to quantify their survival advantages.

## 1. Introduction

Glioblastoma remains the most aggressive and deadly form of primary brain tumor, with average survival rates ranging from 7.8 to 23.4 months after diagnosis [1]. Maximal surgical resection followed by radiation and temozolomide have become standard of care [2,3]. Other treatment options such as carmustine wafers and tumor-treating fields have extended life expectancy by 2.3 months [4] and 4.9 months [5], respectively. Currently ongoing trials are investigating immunotherapy and gene-target therapy to prolong life expectancy [6,7], but 5-year survival remains poor at 5–13% [5].

Following a diagnosis of glioblastoma, patients experience a range of physical, mental, and emotional changes that drastically impact their independence, level of functioning, and quality of life [8]. Many patients and their families seek to gain some control over their disease by making adjustments to their daily health habits, often prompting questioning of the neurosurgical and oncology care teams as to which health practices might have the most benefit. Therefore, we seek to review the existing literature surrounding non-surgical, patient-controlled methods to improve quality of life and prolong survival in glioblastoma.

## 2. Materials and Methods

A review of the English language literature was conducted between the period of September 2020 through to February 2021 through PubMed and Google Scholar using terminology including, but not limited to “diet and glioblastoma survival”, “exercise and glioblastoma survival”, “vitamins and glioblastoma survival”, “supplements and glioblastoma survival”, “caffeine and glioblastoma survival”, “alcohol and glioblastoma survival”, “cannabis and glioblastoma survival”, “nicotine and glioblastoma survival” and variations of these terms. We also searched for current clinical trials involving the listed topics. Reference lists were scanned for potential additional resources. No publication type or date restrictions were applied to the search. Accepted study designs included experimental studies, observational studies, and animal studies. Review articles were examined to be able to retrieve the primary source references. Included articles were those that evaluated the impact of the intervention on glioblastoma cell or patient survival or quality of life.

## 3. Results

### 3.1. Diet

Hyperglycemia has been shown to increase the prevalence and promote the proliferation of a range of cancers [9]. Glucose has been shown to be the primary energy source for high-grade gliomas even in the presence of adequate oxygen [10], and glioblastoma cells grown in high-glucose media replicate significantly faster than those grown in normal-glucose media by up-regulating growth factor receptors [11]. In fact, patients with higher blood glucose levels during treatment had worse overall survival by several months compared to those with normal glucose levels [12,13]. Similarly, patients with pre-existing type 2 diabetes mellitus have significantly shorter progression-free survival and overall survival compared to non-diabetics [14]. In addition, patients with body mass index (BMI) > 30 had a decreased progression-free survival compared to those with BMI < 25 [14]. Therefore, one might reasonably conclude that dietary control or restriction could impact patient outcomes after glioblastoma diagnosis. The ketogenic diet (KD), which is composed of a high fat:carbohydrate ratio, has proven effective in managing other neurologic conditions such as childhood epilepsy [15]. The theory of its success relates to the brain’s use of ketone bodies for energy rather than glucose which induces metabolic changes that protect normal brain cells [16]. A number of studies looking at the effects of the KD on malignant glioma in mouse models have confirmed that the KD slows tumor growth and increases survival due to changes in immune response, gene expression, and amount of reactive oxygen species [17,18,19,20,21]. When combined with radiation, the KD has an even greater survival advantage compared to the KD alone, a standard diet, and a standard diet with radiation [21]. Human studies to date have focused on the safety and feasibility of maintaining the KD during glioblastoma treatment in small sample sizes [22,23,24], while the reports discussing outcomes have been mostly anecdotal. Of the two case reports involving glioblastoma in adults, one patient experienced dramatic and unexpected neuroimaging improvements while on the KD [25], and another patient who maintained the KD after diagnosis had no clinical or radiographic indications of recurrence at 24 months post-operatively [26]. In both patients, no or limited steroid medication was used. Benefits in two pediatric patients with malignant glioma have been reported as well [27]. A recently published prospective randomized trial evaluating the efficacy of dietary changes on re-irradiation in 44 patients with recurrent glioblastoma found that patients in the KD–intermittent fasting (KD–IF) group who achieved blood glucose levels below 83.5 mg/dL had significantly longer progression-free survival (PFS) and overall survival (OS) compared to those who achieved glucose levels above 83.5 mg/dL; however, there was no overall difference in PFS or OS when comparing patients receiving the standard diet with all patients in the KD–IF group [28]. Currently, there are multiple clinical trials investigating the effects of the ketogenic diet on glioblastoma (NCT03278249, NCT01865162, NCT02302235, NCT04691960, NCT03451799, NCT04730869).

Calorie restriction (CR), or short-term fasting, is another dietary modification that could have an impact on glioma survival. Calorie restriction has been shown to extend life span in yeast, mice, and primates, and it selectively protects normal cells over cancer cells from chemotherapy through reduced reactive oxygen species, changes insulin-like growth factor (IGF-1) signaling, and other mechanisms that enhance stress response of normal cells [29,30,31]. When tested in mice with glioma, those who were exposed to periods of short-term starvation (STS) showed slowed progression of the glioma and a longer overall survival, both alone and particularly when combined with temozolomide, compared to control mice [32]. Other studies have confirmed that CR rather than STS reduces tumor invasiveness and protects against chemotoxicity in mice with glioma [33,34]. While there are no human studies evaluating the effects of CR or STS on glioblastoma survival, two clinical trials related to this topic are currently in progress (NCT04461938, NCT02286167).

### 3.2. Exercise

Exercise has a range of benefits for the brain, including growth and development, promotion of brain plasticity, mood elevation and stress relief, and treatment of neurologic disorders such as Alzheimer’s Disease and multiple sclerosis [35]. In glioblastoma, physical activity has been shown in mice to significantly reduce tumor cell proliferation, delay motor deterioration, and support the ability for self-care [36]. When combined with temozolomide, exercise further reduces tumor size in mice compared to untreated mice and those treated with temozolomide alone [37]. In humans with glioma, implementation of a regimented exercise program improves motor function and cognitive functioning and reduces depression, anxiety, headaches, fatigue, and communication deficits compared to those who do not participate [38,39,40]. One study even showed that exercise behavior in patients with grades III and IV glioma was an independent predictor of survival, with those participating in >9 metabolic-equivalent hours/week having a median survival of 7.8 months longer than those participating in <9 metabolic-equivalent hours/week of exercise [41]. In pediatric brain tumor patients treated with a minimum of surgery and cranial radiation, a 12-week exercise regimen resulted in increased cortical thickness primarily in sensorimotor regions and increased white matter volume corresponding with improvements in physical, behavioral, and cognitive functioning [42]. A similar study showed improved bilateral coordination in the exercise group which was maintained for 12 weeks after conclusion of the exercise program [43]. Currently, an ongoing clinical trial is looking at improvement in progression-free survival, overall survival, and quality of life in glioblastoma patients participating in exercise regimens (NCT03390569), while another is evaluating the influence of physical activity on quality of life and well-being in patients with high-grade glioma (NCT03775369).

### 3.3. Vitamin Supplementation

A survey published in 2010 among glioma patients in Germany found that 40.3% of 621 patients used some form of complementary therapy as an addition to their standard medical treatment [44]. While this therapy ranged from magic to acupuncture and homeopathy, 31% of those users supplemented their treatment with vitamin use, and the vast majority of patients engaged in this alternative treatment on the advice of friends and relatives rather than medical professionals. A 2015 study from centers around the United States found that 77% of the surveyed glioblastoma patients reported using complementary therapy, with vitamin supplementation making up the majority form of therapy [45].

Forms of vitamin A, particularly retinoic acid, have been shown to inhibit proliferation of tumor cells and stem-cell marker expression among some human glioblastoma cell lines [46,47]. This research has prompted several clinical studies. Two phase II trials involving giving oral cis-retinoic acid (CRA) to recurrent glioblastoma patients in place of other treatments showed the drug was well tolerated and that it had perhaps some stabilizing effect on the tumor with improved progression-free survival at 6 months [48,49]. A similar retrospective study in a similar patient population showed modest results in response to CRA compared to other chemotherapeutic options [50]. A third phase II trial evaluating oral all-trans-retinoic acid use in recurrent glioblastoma showed no significant effect on tumor progression or survival [51]. Of patients receiving CRA along with temozolomide during the initial treatment of glioblastoma, none had improved progression-free or overall survival compared to those receiving temozolomide alone [52]. There is one ongoing clinical trial evaluating whether isotretinoin can help control glioblastoma when given with temozolomide (NCT00555399).

Vitamin C use as a form of cancer treatment was first proposed in 1976 [53], and two randomized clinical trials utilizing an intravenous (IV) form of the vitamin have shown promising survival results for patients with ovarian cancer and acute myeloid leukemia [54,55]. Additionally, an IV form of vitamin supplementation seems more effective than oral supplementation [56]. In preclinical trials in glioblastoma, ascorbic acid altered intracellular iron metabolism, causing toxic effects on the cancerous cells compared to normal human astrocytes and sensitized the glioblastoma cells to chemotherapy and radiation [57]. These results prompted a phase I clinical trial showing that IV vitamin C was tolerable and safe to use in combination with standard treatment for patients with glioblastoma [57]. One anecdotal report describes a 55-year-old patient who received 46 months of IV vitamin C in combination with standard treatment for glioblastoma and survived more than 4 years after diagnosis [58]. Currently, there are both active phase I and phase II clinical trials evaluating high-dose IV ascorbic acid use in glioblastoma patients in addition to standard treatments (NCT01752491, NCT02344355).

Vitamin D receptors have been identified at significantly higher levels in glioblastoma compared to lower-grade gliomas, suggesting these as a target for treatment [59]. Multiple in vitro studies have demonstrated the cytotoxic, antiproliferative, and antimigration effects of vitamin D metabolites on human glioma cell lines [60,61,62,63,64]. Glioblastoma patients with blood vitamin D levels greater than 30 ng/mL prior to initiation of chemotherapy and radiation have longer overall survival [65], and those who reported vitamin D use after diagnosis of glioblastoma have been reported to have a survival advantage [45]. However, not all glioblastoma cell lines express vitamin D receptors (VDR) [66,67], and patients with positive VDR expression have longer overall survival [67]. A phase II clinical study of alfacalcidiol administration during treatment for glioblastoma demonstrated safety of the supplementation as well as an unexpectedly significant clinical and radiographic regression of the tumors in 2 of the 10 patients and median survival of 21 months [68]. To date, there are no current ongoing clinical trials involving the use of vitamin D in any form in the treatment of glioblastoma.

Vitamin B_3_ has shown promising results in controlling tumor growth and prolonging survival in rat glioma studies [69,70]; however, no human studies have been published to date. A phase I/II clinical trial evaluating niacin in glioblastoma is in the early stages of enrollment (NCT04677049). Similarly, vitamin E in the form of tocopherols has been shown to inhibit tumor cell proliferation in in vitro rat and human glioma studies [71,72]. An observational study linked vitamin E use in glioblastoma patients with a non-significantly higher mortality compared to nonusers [45], although another did not find a significant impact of vitamin E intake on survival, even at higher doses [73]. No prospective or ongoing clinical trials involve vitamin E use in the treatment of glioblastoma.

While most studies have evaluated the addition of vitamins to a treatment regimen, some data suggest that vitamin B12 deficiency is beneficial for cancer patients because of the crucial role that cobalamin plays in DNA replication. A recent in vitro study found that using a cobalamin antagonist prevents glioblastoma cell growth without inducing harm or mortality to developing healthy zebrafish embryos [74]. Elevated cobalamin levels have been linked to shorter overall survival times in multiple types of cancer including lung, hepatobiliary, colorectal, breast, head and neck, and “other tumor groups”, although glioblastoma was not specifically included [75].

### 3.4. Other Dietary Supplements

Melatonin, an antioxidant produced by the pineal gland, has been shown to have an anticancer effect on a range of cancer types [76]. A recent review highlighted the antineoplastic effects of melatonin in glioblastoma both in vitro and in vivo [77], and melatonin has even been shown to have a synergistic effect with temozolomide in human glioblastoma cells [65]. Only one human clinical trial evaluated melatonin as a treatment adjunct, and while this study was completed prior to the adoption of the current standard of care, patients who were treated with radiation plus oral melatonin had a significantly higher one-year survival rate than those who received radiation alone [78].

Several other over-the-counter supplements and medications including lipoic acid, resveratrol, caffeine, curcumin, acetaminophen, aspirin, and non-steroidal anti-inflammatory drugs (NSAIDs) have shown promising results in controlling glioblastoma in in vivo and/or in vitro studies, but none have been studied in human populations or clinical trials [67,79,80,81,82,83,84].

### 3.5. Caffeine

Caffeine, a neurostimulant capable of penetrating the blood–brain barrier, induces a number of effects on the brain including increasing energy metabolism and decreasing cerebral blood flow [85,86]. The consumption of caffeinated beverages including coffee and tea has been associated with a lower risk of glioma in some epidemiological studies [87,88,89], although caffeinated soft drinks did not show the same statistically significant association [87,88]. In in vitro studies, caffeine has been shown to inhibit human glioma cell proliferation, invasion, and migration [90,91,92,93,94]. In vivo, caffeine inhibits cell proliferation, induces apoptosis, reduces tumor volume, and increases survival at significant rates [90,93]. When combined with chemoradiation agents, caffeine augments the effects of cisplatin and temozolomide on human glioma cells [80,95] and radiosensitizes human glioma cells, particularly those with *PTEN* mutations [96]. Only one prospective human trial from 1987 used caffeine as a therapeutic agent in glioma treatment, although caffeine-induced seizures prevented dose escalation to a level expected to potentiate the cytotoxicity of the involved chemotherapeutic agents [97]. Surprisingly, no other human studies have been performed, and there are no active clinical trials. A recent review highlights in detail the promising future role for methylxanthines including caffeine in the treatment of glioblastoma [98].

### 3.6. Alcohol

Another commonly consumed beverage, alcohol is considered to be a strong risk factor for a number of non-neurologic cancers [99]; however, two meta-analyses did not show a convincing association between alcohol consumption and glioma risk [100,101]. Resveratrol, a polyphenol found in grapes, has been shown in in vitro and in vivo studies to have antineoplastic potential [79], but no human studies or clinical trials have investigated its use or the use of alcohol in the treatment of glioblastoma.

### 3.7. Cannabis

With the increasing legality of marijuana use for both recreational and medical purposes, more patients have expressed interest in adding this to their treatment regimen. Two recent reviews have described in detail the anti-inflammatory, antioxidant, and neuroprotective effects of cannabinoids in the CNS and the relation of cannabinoid receptor type 1 (CB1R) and cannabinoid receptor type 2 (CB2R) expression in glioma to treatment possibilities [102,103]. Cannabinoids, for example, can protect astrocytes and oligodendrocyte progenitors from apoptosis in vitro [104,105]; they also play an integral role in the immune system, particularly CB2R, suggesting a role for immunotherapy in glioblastoma treatment [102]. Cannabidiol (CBD), a non-psychoactive cannabinoid, inhibits growth, activates apoptotic pathways, and induces oxidative stress in human glioma cells both in vitro and in vivo [106,107]. Combining Δ^9^-tetrahydrocannabinol (THC) with CBD has an even greater antiproliferative effect when used in conjunction with temozolomide, more so than temozolomide with or without CBD [108]. Another alternative to THC includes ajulemic acid, a synthetic non-psychoactive cannabinoid, which also inhibits growth of human glioma cells both in vitro and in vivo [109].

Regardless of the potential benefits on restricting tumor growth, a third of surveyed glioblastoma patients reported using some form of medical marijuana for relief from symptoms ranging from pain and nausea to insomnia and emotional lability, with the vast majority of those reporting symptom improvement [110]. Similarly, over 67% of patients with primary brain malignancy reported a moderate to significant improvement in their overall condition after 6 months of cannabis use [111]. A phase II randomized clinical trial confirmed that daily use of THC was both tolerable and beneficial to the overall wellbeing and quality of life for the patient [112]. A phase I study has even demonstrated safety in administering THC intracranially through a subcutaneous port [113]. This study also demonstrated that the tumors treated with THC had fewer viable cells and reduced tumor cell proliferation [113]. Regarding survival, patients with recurrent glioblastoma who received a 1:1 THC:CBD dose in addition to temozolomide had a six month longer median survival compared to those receiving a placebo in addition to temozolomide in a small blinded, randomized trial [114]. A survival advantage was also found at both 1 and 2 years among glioblastoma patients who ingested 50mg of cannabinoid oil daily in addition to standard treatments compared to those with low or palliative cannabis use [115]. Currently, a clinical trial investigating the THC/CBD combination with temozolomide and radiation is ongoing in patients with newly diagnosed glioblastoma (NCT03529448).

### 3.8. Cigarette Smoking

While cigarette smoking has not clearly been shown to be a risk factor for development of glioblastoma, a couple studies have demonstrated an increased rate of malignant glioma in patients with a smoking history, particularly for heavy smokers [116,117]. As an addictive habit that is often used as a source for stress relief, about 64% of all cancer patients continue to smoke after diagnosis [118]. Nicotine, while not a carcinogen itself, can have a number of pro-tumor effects on glioblastoma cells including promoting tumor cell growth and protecting against chemotherapy and radiation [119]. In fact, one study demonstrated that tobacco use in glioblastoma patients with Karnofsky performance scores < 70 had worse outcomes than nonsmokers [120]. Unfortunately, there are limited human studies evaluating the effect of smoking on overall survival in glioblastoma patients, and there are no related ongoing clinical trials to date.

### 3.9. Additional Studies

A couple of studies have evaluated the use of complementary therapy and/or alternative treatments in glioblastoma [45,121]. While both studies focus heavily on a variety of dietary supplements, some of which are discussed here, one does include meditation, yoga, and prayer, none of which improved quality of life by statistically significant differences [121].

Other factors that have been associated with longer overall survival in glioblastoma are participation in clinical trials and being married [122,123]. The five-year survival rate for widowed patients with glioblastoma is 6.2% compared to 13.8% in married patients [124], suggesting that widowed patients may benefit from more access to resources providing psychological support. Additionally, unmarried patients are more likely to present with larger tumors, are less likely to have aggressive treatment, and have shorter survival after diagnosis compared to married patients when adjusted for treatment and prognostic markers [125]. Finally, despite the promising results of vitamin D supplementation, amount of sunlight exposure has not been found to impact glioblastoma survival [126].

Table 1 summarizes the highest level of evidence in human studies for the effect of lifestyle modifications on glioblastoma survival.

Many of the above lifestyle changes are habits that are generally associated with overall health and wellbeing. While one could argue that everyone should adopt these habits to promote health and lengthen lifespan, only a portion of people are motivated to follow current diet and exercise recommendations. Those faced with a newly diagnosed terminal illness may have a renewed interest in health and wellness, and they may be eager to make such changes in their lifestyles. Many trials discussed here are in vitro studies, and while these are important first steps toward advancements in glioblastoma care, they do not always translate into effective in vivo studies due to difficulty identifying an ingestible agent, avoiding first-pass liver effects, achieving therapeutic drug levels, and penetrating the blood–brain barrier. Human clinical trials are necessary to determine which health habits in particular are beneficial in glioblastoma patients in conjunction with today’s standard of care treatments.

## 4. Conclusions

In glioblastoma patients who are motivated to make lifestyle adjustments to improve their outcomes, dietary restriction of sugar and caloric intake in particular seems to have the most promise, with exercise, vitamin supplementation, and cannabis use showing potential benefits as well. Ongoing human clinical trials are necessary to determine the true impact of these health habits on survival in conjunction with today’s standard of care treatments of glioblastoma. We anticipate that the best outcomes in glioblastoma will be through a combination of surgical, medical, and lifestyle-based approaches. 

## Figures and Tables

**Table 1 brainsci-11-00533-t001:** Highest level of evidence in human studies of lifestyle modification on glioblastoma survival.

Category	Type of Study	Author	Findings	Current Clinical Trial	Current Status; Estimated Date of Completion
Ketogenic Diet	Prospective randomized clinical trial	Voss et al. [28]	Patients in the ketogenic diet–intermittent fasting group receiving reirradiation for GBM who achieved blood glucose levels < 83.5 mg/dL had significantly longer PFS and OS	Ketogenic diet in combination with standard-of-care radiation and temozolomide for patients with glioblastoma (NCT03451799)	Recruiting; April 2021
				Feasibility study of modified Atkins ketogenic diet in the treatment of newly diagnosed malignant glioma (NCT03278249)	Recruiting; January 2021
				Feasibility, safety, and efficacy of a metabolic therapy program in conjunction with standard treatment for glioblastoma multiforme (NCT04730869)	Not yet recruiting; November 2022
Calorie Restriction	No human trials; mouse studies only	N/A	N/A	Glioma modified Atkins-based diet in patients with glioblastoma (GLAD) (NCT02286167)	Completed with no results posted yet; July 2019
				Characterization of metabolic changes in the glioma tumor tissue induced by transient fasting (ERGO3) (NCT04461938)	Recruiting; March 2022
Exercise	Prospective observational cohort study	Ruden et al. [41]	Exercise behavior is an independent predictor of survival with those exercising ≥ 9 MET-h/wk having a median survival of 21.84 months compared to 13.03 months in those exercising < 9 MET-h/wk	Does exercise improve progression-free survival in glioblastoma? A prospective single arm intervention trial (NCT03390569)	Recruiting; December 2020
				Influence of physical activity in patients with high grade glioma on patients’ psychological well being, sleep and quality of life (NCT03775369)	Recruiting; December 2021
Vitamin A	Multi-center phase II clinical trial	Jaeckle et al. [49]	Patients with recurrent glioblastoma receiving cis-retinoid acid and temozolomide exceeded the goal of 20% increase in PFS at 6 months compared to historical controls	Phase I/II adaptive randomized trial of vorinostat, isotretinoin, and temozolomide in adults with recurrent glioblastoma multiforme (NCT00555399)	Active, not recruiting; November 2024
Vitamin B_3_	No human studies	N/A	N/A	A phase I/II study of niacin in patients with newly diagnosed glioblastoma receiving concurrent radiotherapy and temozolomide followed by monthly temozolomide (NCT04677049)	Recruiting; January 2026
Vitamin C	Phase 1 clinical trial	Schoenfeld et al. [57]	High-dose ascorbic acid is safe and well-tolerated in patients with glioblastoma with a trend toward longer PFS and overall survival compared to historical averages	Pharmacological ascorbate combined with radiation and temozolomide in glioblastoma multiforme: A Phase 2 trial (NCT02344355)	Active, not recruiting; December 2024
				A phase I trial of high-dose ascorbate in glioblastoma multiforme (NCT01752491)	Active, not recruiting; December 2021
Vitamin D	Phase 2 clinical trial	Trouillas et al. [68]	Oral alfacalcidol is safe. 20% of glioblastoma patients had significant clinical and radiographic response and were alive >4 years after diagnosis	None	N/A
Melatonin	Randomized controlled trial	Lissoni et al. [78]	42.9% of glioblastoma patients receiving oral melatonin in addition to radiotherapy were alive at one year compared to 6.2% of those receiving radiotherapy alone	None	N/A
Caffeine	Phase 2 clinical trial	Stewart et al. [98]	Caffeine-induced seizures prevented dose escalation to a therapeutic level	None	N/A
Alcohol	No human studies	N/A	N/A	None	N/A
Cannabis	Randomized double-blind placebo-controlled trial	Twelves et al. [114]	1-year survival in patients receiving CBD:THC with temozolomide was 83% compared to 56% in those receiving temozolomide plus placebo	Phase Ib, open-label, multicenter, intrapatient dose-escalation clinical trial to assess the safety profile of the TN-TC11G (THC + CBD) combination with temozolomide and radiotherapy in patients with newly diagnosed glioblastoma (NCT03529448)	Not yet recruiting; June 2023
Smoking Cessation	Retrospective cohort study	Paravati et al. [120]	Smokers with KPS < 70 have median survival of 5.2 months compared to 8 months in nonsmokers with KPS < 70	None	N/A
Marital Status	Retrospective population-based cohort study	Xie et al. [123]	Longer overall survival for married patients	None	N/A

Abbreviation: Karnofsky performance scale (KPS); progression-free survival (PFS); metabolic-equivalent hours per week (MET-h/wk).

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
