# Peer review of "Daily Lifestyle Modifications to Improve Quality of Life and Survival in Glioblastoma: A Review"

_brainsci, 2021, doi:10.3390/brainsci11050533_

Round 1

Reviewer 1 Report

Review summary:

In this article titled “Daily Lifestyle Modifications to Improve Quality of Life and Survival in Glioblastoma: A Review” authors have reviewed published literature focusing on the quality of life in patients with GBM. I have the following concerns regarding this manuscript.

  1. Authors have mentioned few ongoing clinical trials in this study. It would be beneficial to add more about these trials in terms of what is being done and when the results are expected.
  2. The authors have mainly focused on dietary lifestyle changes in this paper. I believe the title should be changed to reflect that and other commonly used beverages such as alcohol, tea/coffee etc should be mentioned in the study.
  3. Authors should include a table to summarize major studies for each of the factors evaluated.

There are several papers published on quality of life in patients with glioblastoma. A review of all factors is almost impossible and exhaustive. However, the authors should include more dietary factors as suggested.

Author Response

  1. We addended the table to include the current status of the clinical trials and the expected date of completion.
  2. We added sections addressing caffeine and alcohol as suggested. We disagree that a change to the title of the manuscript is appropriate given that we discuss non-dietary factors as well.
  3. Given that the existing table already addresses the most significant human trial for each section of the manuscript, we feel additional tables will be redundant and detract from the rest of the paper.

Reviewer 2 Report

The manuscript “Daily Lifestyle Modifications to Improve Quality of Life and Survival in Glioblastoma: A Review.” by Sarah Travers and Scott Litofsky ia a well written narrative review of an complex field that is of great importance in Neuro-oncology daily practice.

Please complete the references relative to the effects of Vitamin D introducing the following citations:

  1. Magrassi L, Bono F, Milanesi G, Butti G. Vitamin D receptor expression in human brain tumors. J Neurosurg Sci. 1992 Jan-Mar;36(1):27-30. PMID: 1323646.
  2. Magrassi L, Butti G, Pezzotta S, Infuso L, Milanesi G. Effects of vitamin D and retinoic acid on human glioblastoma cell lines. Acta Neurochir (Wien). 1995;133(3-4):184-90. doi: 10.1007/BF01420072. PMID: 8748764.
  3. Magrassi L, Adorni L, Montorfano G, Rapelli S, Butti G, Berra B, Milanesi G. Vitamin D metabolites activate the sphingomyelin pathway and induce death of glioblastoma cells. Acta Neurochir (Wien). 1998;140(7):707-13; discussion 713-4. doi: 10.1007/s007010050166. PMID: 9781285.

Author Response

  1. We included and referenced the three papers on vitamin D as suggested.

Reviewer 3 Report

For the most part, this is a well-presented and well-written review. The information provided here is useful for clinicians and doctors to improve on the current treatment regimen for individuals with glioblastoma, especially sections on ketogenic diet, vitamins and cannabis. The interpretation based on clinical findings is also well-explained. However, there are aspects that need to be improved upon- some inaccuracies in references, and also on making broad conclusions based on solely in vitro findings.

  • Line 214-216, Citations 84-86- you mention about “cannabinoids selectively targeting brain tumor cells, while sparing and even promoting the survival of healthy brain cells”. This is completely false and misleading with the references you provided. The references you provided shows that cannabinoids protect astrocytes from ceramide-induced apoptosis, and the other study shows promotion of oligodendrocyte progenitor cell survival. And the other reference is a review on cannabinoids and neuroprotection. Cannabinoids have no such ability to discriminate between cancer and healthy brain cells, while only targeting the former. Please refrain from making inaccurate conclusions from in vitro studies that have employed only primary cell culture models.
  • Please also note that there are excellent recent reviews on cannabinoids and cancer biology by MDPI to help elaborate on mechanisms of cannabinoids (Moreno et al., 2020, Cancers-MDPI)
  • It would help to relate some of your findings to astroglial functions considering they are the pivotal cell type to be transformed in glioblastoma. While some information is provided on this aspect in the vitamin and cannabis sections, there is not enough information that is presented in your review.
  • Both astroglial CB1R and CB2R play major roles in regulating neuroinflammatory states (Haspula and Clark., 2020, IJMS-MDPI). Considering that immunotherapy has a pivotal place in the treatment of glioblastoma, the section on in vitro and in vivo effects of cannabis/cannabinoids on astrocytes needs to be elaborated to strengthen this section.
  • What about antioxidant supplementation for glioblastoma (Abdollahi, 2013, Toxicol Appl Pharmacol)
  • In various parts of the review you mention about the utility of various therapies in both “in vitro and in vivo studies”. In vivo studies reporting efficacy in various cancers is of much greater significance than in vitro findings using commercially available cancer cell lines. You need to make a distinction here since findings reported under in vitro settings need to be translated to in vivo studies before making any prominent conclusions. You can definitely mention about in vitro data and possible mechanisms of the therapy. But if there are reports of just in vitro findings for certain therapies, such as vitamin D metabolites (58-60) and vitamin E (67,68), you have to let the readers know about the potential shortcomings of in vitro studies.  

Author Response

  1. We revised and augmented the current clinical trials in the body of the manuscript and in the table
  2. We augmented some of the discussion in the diet section
  3. We renumbered all of the references to reflect the new additions